# Effect of Adhesion Conditions on the Shear Bond Strength of 3D Printing Resins after Thermocycling Used for Definitive Prosthesis

**DOI:** 10.3390/polym15061390

**Published:** 2023-03-10

**Authors:** You-Jung Kang, Yeseul Park, Yooseok Shin, Jee-Hwan Kim

**Affiliations:** 1Department of Prosthodontics, Oral Science Research Center, College of Dentistry, Yonsei University, Seoul 03722, Republic of Korea; 2Department of Conservative Dentistry, Oral Science Research Center, College of Dentistry, Yonsei University, Seoul 03722, Republic of Korea

**Keywords:** 3D printing resin, shear bond strength, surface roughness, adhesion conditions, surface treatment, thermocycling

## Abstract

Three-dimensional (3D) printing polymers such as urethane dimethacrylate (UDMA) and ethoxylated bisphenol A dimethacrylate (Bis-EMA) are typically used in definitive prosthesis and require surface treatments before bonding. However, surface treatment and adhesion conditions often affect long-term use. Herein, polymers were divided into Groups 1 and 2 for the UDMA and Bis-EMA components, respectively. The shear bond strength (SBS) between two types of 3D printing resins and resin cements was measured using Rely X Ultimate Cement and Rely X U200, according to adhesion conditions such as single bond universal (SBU) and airborne-particle abrasion (APA) treatments. Thermocycling was performed to evaluate the long-term stability. Sample surface changes were observed using a scanning electron microscope and surface roughness measuring instrument. The effect of interaction between the resin material and adhesion conditions on the SBS was analyzed via a two-way analysis of variance. The optimal adhesion condition for Group 1 was achieved when U200 was used after APA and SBU, whereas Group 2 was not significantly affected by the adhesion conditions. After thermocycling, the SBS significantly decreased in Group 1 without APA treatment and in the entire Group 2. Additionally, porosity, along with increased roughness, was observed on both material surfaces after APA.

## 1. Introduction

Several three-dimensional (3D) printing technologies and materials have been developed with advances in computer-aided design/computer-aided manufacturing (CAD/CAM) technology. To manufacture various restorations in the dental field, 3D printing or prototyping is typically used to fabricate 3D models by layering polymer materials [1,2,3,4]. Notably, 3D printing technology can easily and accurately manufacture detailed and complex prostheses. This approach significantly reduces the time and labor required in clinical and laboratory procedures compared to conventional methods [1,5,6,7,8]. Given these advantages, 3D printing resin materials for long-term definitive prostheses have been developed [9,10], and research on 3D printing materials is underway [2,11,12].

The primary component of 3D printing resins used for definitive prosthesis is either urethane dimethacrylate (UDMA) or ethoxylated bisphenol A dimethacrylate (Bis-EMA). When a new dental material is developed, various factors, such as the mechanical properties determined by the composition of the material, adhesion to existing dental cement, and processing method, must be considered. Among these, adhesion is a clinically essential factor in ensuring a successful restorative treatment. Certain previous studies have reported on the shear bond strength (SBS) of 3D printing resin materials for temporary teeth with provisional repair materials [8,13,14]. However, no studies have been conducted on the adhesion between 3D printing resins used as definitive prostheses and resin cement. Thus, research on proper adhesion methods is essential.

Typically, surface treatments are essential to increase the bonding strength between a restoration and a tooth. Accordingly, studies have evaluated several dental restorative materials combined with various surface treatment methods [14,15]. In particular, methods to increase the bonding strength of restorations include mechanical and chemical surface treatments; notably, grinding, sandblasting, laser-based methods, and plasma-based approaches are methods suggested for mechanical treatments [16,17]. Among these, airborne-particle abrasion (APA) can increase adhesion by creating a porous structure and increasing surface roughness [15,18]. Chemical surface treatment methods include etching with sulfuric acid and a piranha solution, applying adhesive primers and silane coupling [17,19], and bonding using adhesive primers or bonding agents that can increase the SBS between materials and cement. Notably, chemical bonding between methacrylate monomers is widely known as the bonding mechanism between resin cement and resin-based restorative materials. A recently developed universal adhesive contains 10-methacryloyloxydecyl dihydrogen phosphate (10-MDP), a methacrylate-based monomer that can improve the bonding between materials and cement. Bonding agents with MDP monomers have been reported to have a higher SBS; thus, combining these bonding agents with mechanical surface treatments may prove advantageous, particularly for the adhesion of CAD/CAM materials [20,21,22].

Notably, to maintain a long-term high bonding strength with cement under various oral environmental conditions, investigating appropriate surface treatments for 3D printing resin materials used as the definitive prosthesis is clinically meaningful. Therefore, this study aimed to identify appropriate adhesion conditions by analyzing the SBSs of two types of 3D printing polymer materials used as definitive prostheses with resin cement based on the surface treatment method. The null hypothesis of this study is that the SBS does not change by varying the materials or adhesion conditions.

## 2. Materials and Methods

### 2.1. Preparation of Specimens

A process flow diagram of the methodology adopted is presented in Figure 1. Two types of 3D-printable definitive resin materials were used in this in vitro study, wherein Group 1 was based on the UDMA oligomer (TeraHarz TC-80, Graphy, Seoul, Republic of Korea), and Group 2 was based on Bis-EMA-filled dental glass (Permanent Crown Resin, Formlabs, Somerville, MA, USA). All the specimens used in this study were designed with the following specific dimensions using CAD software (MeshMixer, Autodesk, San Rafael, CA, USA): 7 mm in diameter and 3.5 mm in thickness. The designed specimens were fabricated using a 3D printer under the conditions suggested by the manufacturer. Per the recommendation of the manufacturer, disc-shaped resin specimens of Group 1 were printed using a digital light processing (DLP) 3D printer (Sprint Ray Pro 95, Sprint Ray, Inc., Los Angeles, CA, USA). The printed specimens were ultrasonically cleaned using 90% isopropyl alcohol for 10 min to remove any excess resin and then dried. As per the recommendation of the manufacturer, the samples were then cured for 20 min using a curing machine (Cure-M 102H, Graphy, Seoul, Republic of Korea).

A stereolithography (SLA) 3D printer (Form 3, Formlabs, Somerville, MA, USA) was used to print the specimens of Group 2, as recommended by the manufacturer. Likewise, these printed samples were ultrasonically cleaned using 90% isopropyl alcohol for 10 min to remove any excess resin. The completely dried specimens were photocured for 30 min at 60 °C using a curing machine (Formcure, Formlabs, Somerville, MA, USA), as suggested by the manufacturer.

For the SBS test, an adhesive bonding agent (single bond universal (SBU), 3M ESPE, St. Paul, MN, USA) and the following two different types of resin cement were used: Rely X U200 and Rely X Ultimate, both obtained from 3M ESPE. All the printed resin specimens were embedded in a resin mold with their surfaces exposed for cement bonding. Additional details on these materials can be found in Table 1. The upper surface of each specimen was polished using 1000-grit silicon carbon abrasive papers at 300 rpm to obtain a uniform surface. This was achieved using a polishing machine under tap water irrigation to eliminate contaminants. The specimens were divided into two types based on the resin material. Each type of resin specimen was then further classified into six groups (with 15 specimens per group) based on the surface treatment method, presence or absence of bonding agent treatment, and type of resin cement. The groups are presented in Table 2.

Furthermore, APA with 50 µm aluminum oxide particles was conducted in a direction vertical to the surface at 0.2 MPa for 10 s at a distance of 10 mm using a sandblaster. After the APA surface treatment, the resin specimens were ultrasonically washed for 5 min to remove the alumina particles from the surface. The bonding agent treatment was performed, as recommended by the manufacturer, by applying SBU to the surface using a micro brush, rubbing for 20 s, and then blowing it with air for 5 s. Following this, the surface was light cured using a 1200 mW LED curing unit (C01 LED Curing Light, Premium Plus) for 10 s. This step was not carried out in the Ultimate groups. The resin cement bonding method was as follows: after resin cement was applied to a plastic mold, it was placed on the resin surface (bonding area of 4.45 mm^2^) and light cured.

For the SBS test, 336 definitive resin specimens of the two types were prepared, with 28 in each group. Half of the specimens from each group were stored in distilled water for 24 ± 2 h at a temperature of 37 °C. The other half of the specimens were subjected to 10,000 thermocycles between 5 and 55 °C in distilled water (70 s per cycle; dwelling time: 30 s, transfer time: 5 s) for aging treatment.

### 2.2. Shear Bond Strength

Each specimen was fixed in a universal testing machine (Bisco, Schaumburg, IL, USA) using a metal jig for SBS evaluation (*n* = 14). The maximum load was measured at a crosshead speed of 0.5 mm/min at 1 mm from the bonding interface until the cement column was dropped. The SBS for each specimen was calculated by dividing the peak load (in N) by the surface area (4.45 mm^2^) to obtain the strength (in MPa). The SBS test was performed according to the International Organization for Standardization (ISO) 10477 [23].

### 2.3. Scanning Electron Microscopy

Surface images of the two types of 3D printing resins were obtained at magnifications of ×500 and ×2000 using scanning electron microscopy (SEM; S-300N, Hitachi, Tokyo, Japan). One representative specimen per group was used, and all specimens were subjected to SEM pre-treatment by applying a platinum coating with a thickness of 100 nm using an ion sputter coater (E-1010, Hitachi).

### 2.4. Roughness

A 3D optical profiler (Contour GT-X3 BASE, Bruker Co., Bremen, Germany) was used to compare and analyze the average surface roughness (Ra) of the 3D printing resins based on the APA surface treatment. One representative sample was used for each group, and the surface roughness and topography were analyzed in nine areas of the sample.

### 2.5. Statistical Analysis

A statistical analysis software (SPSS 25.0, SPSS, Inc., Chicago, IL, USA) was used to perform the statistical analysis. All data values were evaluated for normality based on the Kolmogorov–Smirnov and Shapiro–Wilk tests. A two-way analysis of variance (ANOVA) with Bonferroni post-analysis was performed to confirm the effect of the type of 3D printing resin, adhesion conditions with the resin cement, and their interaction on the SBS values of each material. Similarly, the Mann–Whitney U test was conducted to investigate the effects of thermocycling on the SBS. The Ra values were analyzed to compare the surface treatments for each material via an independent t-test. The significance cutoff in all tests was α = 0.05.

## 3. Results

### 3.1. Shear Bond Strength

The results of the average SBS values and significant differences observed according to the 3D printing resin material and adhesion conditions are presented in Figure 2, Figure 3, Figure 4 and Figure 5. As stated, a two-way ANOVA was conducted to verify the main and interaction effects of each 3D printing resin material and adhesive condition on the SBS; the results indicate that the main effect of the material and the interaction effects of the material and adhesion condition were significant on the SBS before thermocycling; however, the main effect of the adhesion condition was insignificant. After thermocycling, both the materials and adhesion conditions demonstrated significant main and interaction effects on the SBS.

By confirming the interaction effect through Bonferroni’s multiple comparison tests, the average SBS of Group 1 before thermocycling was found to be significantly higher in the ABU group than in the CU group. The ABU group had the highest numerical value at 24.7 ± 4.9 MPa, but no statistically significant difference was observed compared with the other adhesion condition groups AU, CBU, and CBUT. In Group 2, the average SBS was numerically the highest in the ABUT group at 19.4 ± 2.7 MPa, but no significant difference could be noted compared with the other groups. In addition, the SBS of Group 1 was generally higher than that of Group 2, and significant differences were observed compared to the CBU, ABU, and CBUT groups.

After thermocycling, the average SBS of Group 1 was 8.1 ± 3.1 MPa in the CU group, which was significantly lower than that of all groups, and the average SBS of the ABU group was 22.5 ± 3.3 MPa, which was statistically the highest. No significant difference in the average SBS of Group 2 was observed compared to that of the entire group. In addition, the SBS of Group 1 was significantly higher in the AU, ABU, CBUT, and ABUT groups compared to those of Group 2, and it was significantly lower in the CU group.

The results of the significant differences in the SBSs of each material according to thermocycling are presented in Table 3. After thermocycling, the average SBS in Group 1 decreased significantly in the CU, CBU, and CBUT groups without APA treatment compared to the values before thermocycling, and in Group 2, all groups showed a significant decrease.

### 3.2. Scanning Electron Microscopy

Figure 6 and Figure 7 display representative SEM images of both types of 3D printing resins. Compared to the control specimens in each group, APA treatment increased the surface roughness of both materials. In Group 1, microscopic spaces were formed with relative uniformity between the matrices, and in Group 2, irregularly deep and large porous sites were formed, wherein numerous small and fine particles were embedded in the surface or partially exposed. This can be interpreted as the result of exposure of the filler particles of Group 2 after polishing and APA treatment. Note that Group 1 consists of a UDMA-based resin matrix without filler and Group 2 consists of inorganic fillers in a Bis-EMA-based resin matrix. Group 2 specifically contains silane-treated fillers; thus, it can be confirmed that the particles are buried in the matrix.

### 3.3. Roughness

The results of the Ra values according to surface treatments of the two types of 3D printing resins are presented in Table 4, and the representative 3D surface morphology images of each group are presented in Figure 8. Both resins presented significantly increased Ra values compared to those in the control group after APA treatment. In the control group, the Ra values between the two materials showed similar results, but in the APA group, Group 2 showed a significantly higher Ra value than Group 1.

## 4. Discussion

In this study, the SBS was evaluated according to two types of 3D printing resin materials used as definitive prostheses and adhesion conditions, and thermocycling was performed to verify the long-term effect of adhesion. The results of the two-way ANOVA indicated that the SBS of Group 1 tended to be significantly higher than that of Group 2. No significant difference was observed in the SBS according to the adhesion conditions before thermocycling, whereas a significant difference was observed after thermocycling. In addition, the materials and adhesion conditions produced statistically significant interactive effects on the SBS before and after thermocycling. Therefore, the null hypothesis of this study was partially rejected.

As stated, adhesion is a clinically important factor for the long-term use of dental restorations under various oral environmental conditions. Several studies have reported on the SBS for various surface treatment methods in dental materials. APA is generally recommended as a mechanical surface treatment to improve the adhesive strength of CAD/CAM resin restorative materials [24,25]. Previous studies have shown that APA can roughen the surface of resin composites, increase surface energy, and enable cement to penetrate the material, thereby promoting a strong bond between the cement and restorative material and improving micromechanical interlocking [17,26]. In addition, it has been reported that the recommended APA pressure for CAD/CAM resin restorative materials is 0.1–0.2 MPa [27]. Therefore, in this study, as a mechanical surface treatment for 3D printing resins, APA treatment was performed at a pressure of 0.2 MPa using 50 µm alumina particles.

In addition, chemical surface treatment can improve the bonding strength between 3D printing materials and resin cement. One of the main components of the adhesives used for bonding resins is a functional monomer, mainly 10-MDP. The MDP monomer presents better adhesion results when combined with APA [28,29,30]. For this study, single bond universal (SBU), a 10-MDP-based bonding agent, was recommended by the manufacturer. Previous studies have reported that primers or adhesives containing MDP exhibit favorable adhesive performance. Kömürcüoğlu suggested sandblasting treatment and the use of general-purpose adhesives, including MDP, for the surface treatment of CAD/CAM restoration materials to improve the bond strength [20].

SBU also contains Bis-GMA and silane. Several studies have reported that resin composite types of cement containing Bis-GMA act primarily on micromechanical interlocking. In addition, it has been suggested that it is advantageous to use silane treatments or general-purpose adhesives containing silane in the surface treatment to repair CAD/CAM materials [31,32]. It has also been reported that the silane coupling agent is a double molecule that can bind with inorganic (silicon oxide) and organic substances (methacrylate group of resin cement) to achieve proper bonding [30,33,34]. Considering these reports, in this study, a high SBS was reported for the group treated with both APA and SBU. In Group 1, the ABU group showed the highest value at 24.7 ± 4.9 MPa, and the ABUT group showed the highest value at 19.4 ± 2.7 MPa in Group 2. According to a study conducted by Fouquet, using a universal primer containing 10-MDP and silane for a CAD/CAM resin block did not affect the improvement in the SBS. However, it has been reported that the SBS increased significantly when combined with micromechanical surface treatment [35]. Mechanical surface treatment with APA can improve micromechanical retention by increasing the surface area and surface energy of a specimen. It can also maximize the wettability of silanes and resin cements by reducing surface tension. This forms a thin and deep interface when hydrophilic SBU is applied, and after air drying and light curing, SBU changes to hydrophobicity, increasing mechanical and chemical adhesion with resin cement [36]. Therefore, it can be interpreted that a combination of mechanical and chemical surface treatments can significantly improve the bonding strength of 3D printing resins and resin cement.

In this study, an artificial aging process was used to verify the long-term stability of adhesion. Notably, thermocycling is a representative artificial aging method used to investigate dental restorative materials. In this in vitro study, 10,000 thermocycles representing a one-year clinical period were performed at temperatures between 5 and 55 °C to reproduce the oral environment [37]. The SBS of Group 1 before and after thermocycling showed the highest value in the ABU group, whereas no significant difference was observed in the APA-treated group before and after thermocycling. This suggests that the APA treatment is essential for long-term adhesive stability. The SBS of CU in Group 1 was 8.1 ± 3.1 MPa, within the range permitted by ISO 10477 [23], but was significantly lower than that of other groups. Therefore, mechanical treatments and bonding applications are considered necessary for long-term adhesive stability in the oral environment. Group 2 also showed a numerically high tendency in ABUT before and after thermocycling. However, no significant difference was observed in the SBS with a change in adhesive conditions, and a considerable decrease was shown in the SBS before and after thermocycling. The SBS of Group 1 was higher than that of Group 2, which is expected given that UDMA is more polar than Bis-EMA, creating stronger interfacial adhesion with bonding agents and adhesive cement. According to these results, we can confirm the existence of certain preferred adhesion conditions between 3D printing materials and dental cement. It is therefore important to set adhesion conditions according to the materials used.

The effect of the APA treatment on the surface of the 3D printing resin specimens was determined through surface roughness measurements and SEM analysis. The APA treatment resulted in fine irregularities on the surfaces of both materials and improved their roughness. These surface changes improved mechanical bonding with the resin cement by increasing the surface area and wettability. In particular, in Group 2, small and fine particles were observed after the APA treatment. This result agreed with the SEM images acquired for the surface treatment and SBS test for CAD/CAM materials in previous studies [38,39]. Thus, it can be interpreted that the relatively weak resin matrix was partially destroyed by polishing and APA treatment, and the hard inorganic filler remained or was exposed to form a rough surface. The surface roughness after the APA treatment was measured to be 0.79 ± 0.05 µm in Group 1 and 1.00 ± 0.07 µm in Group 2, indicating that the latter possessed a greater surface roughness but lower SBS. This suggests that the non-uniform, deep, and wide porous surface did not mechanically interlock with the resin cement, resulting in reduced long-term stability. Hence, it can be suggested that bonding strength enhancement is affected by the formation of a uniform concave–convex structure and not the increase in surface roughness.

In this study, it was possible to predict the appropriate adhesion conditions and long-term stability for each material by measuring the SBS values using different surface treatments and resin cements for new 3D printing resins. Further research should be conducted on other factors related to the evaluation of SBS and long-term retention according to the use of various primers that reflect the characteristics of the material.

## 5. Conclusions

Based on the findings of this study, the following conclusion could be drawn: The SBS varied depending on the type of 3D printing resin material and adhesion condition. When both materials were treated with APA and SBU, the SBS tended to be high; thus, mechanical and chemical surface treatments must be combined when bonding 3D resin materials. Group 1 presented a high bonding strength with U200 resin cement. Although no significant difference was observed in Group 2, it showed a high bonding strength with the Ultimate resin cement. Group 1 showed high adhesive stability after thermocycling, but Group 2 showed a considerable decrease in the SBS after thermocycling.

Given the existence of preferred adhesion conditions for each material, research on various adhesion conditions between 3D printing materials for definitive prosthesis and dental cement is needed to obtain more clinically meaningful results.

## Figures and Tables

**Figure 1 polymers-15-01390-f001:**
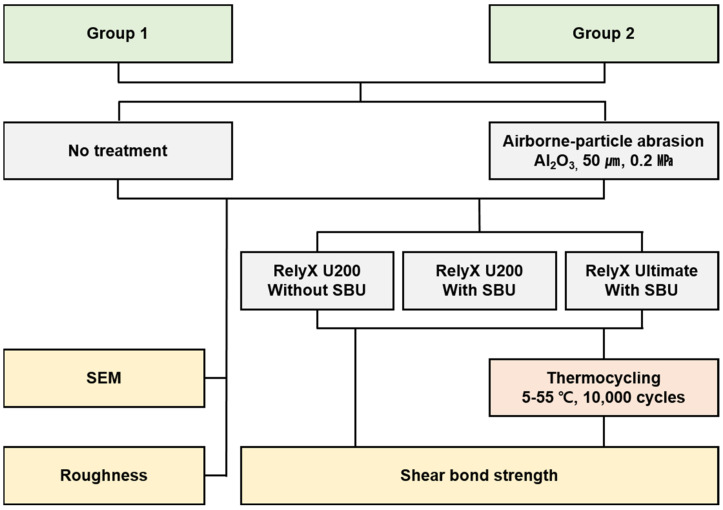
Process flow diagram of the adopted methodology.

**Figure 2 polymers-15-01390-f002:**
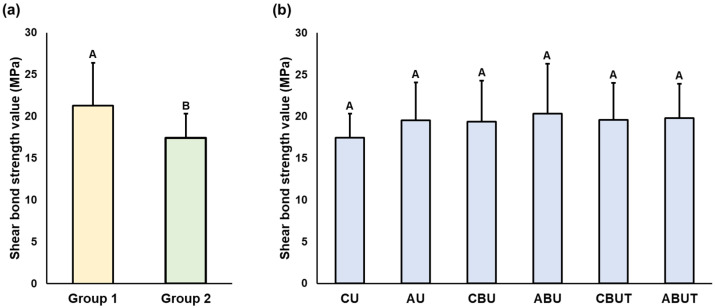
Results of the two-way ANOVAs of shear bond strength (SBS) before thermocycling (**a**) between Groups 1 and 2 and (**b**) between the groups with different adhesion conditions. Uppercase letters indicate significant differences. Data are the means and SD values.

**Figure 3 polymers-15-01390-f003:**
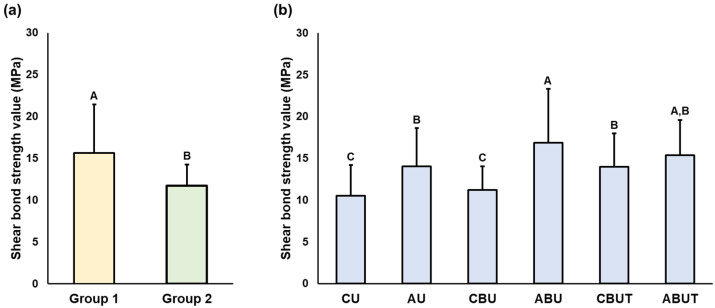
Results of the two-way ANOVAs of SBS after thermocycling (**a**) between Groups 1 and 2 and (**b**) between the groups with different adhesion conditions. Uppercase letters indicate significant differences. Data are the means and SD values.

**Figure 4 polymers-15-01390-f004:**
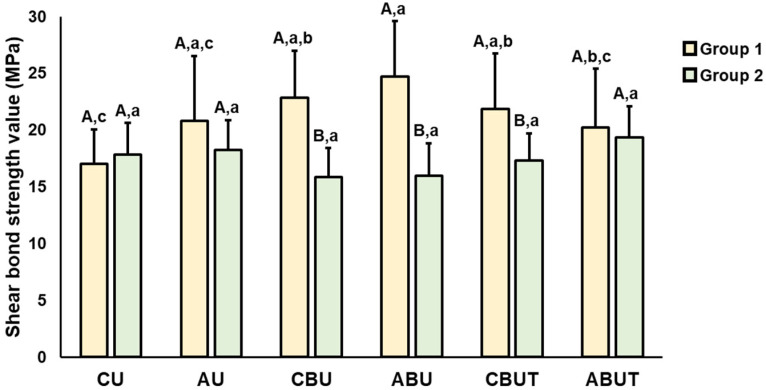
Results of the SBS between Groups 1 and 2 before thermocycling. Uppercase letters indicate significant differences between Groups 1 and 2 within the same adhesion condition, while lowercase letters indicate significant differences between adhesion conditions within each group.

**Figure 5 polymers-15-01390-f005:**
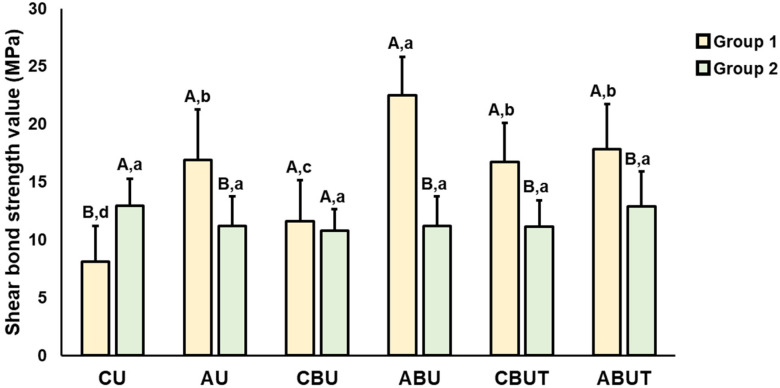
Results of the SBS between Groups 1 and 2 after thermocycling. Uppercase letters indicate significant differences between Groups 1 and 2 within the same adhesion condition, while lowercase letters indicate significant differences between adhesion conditions within each group.

**Figure 6 polymers-15-01390-f006:**
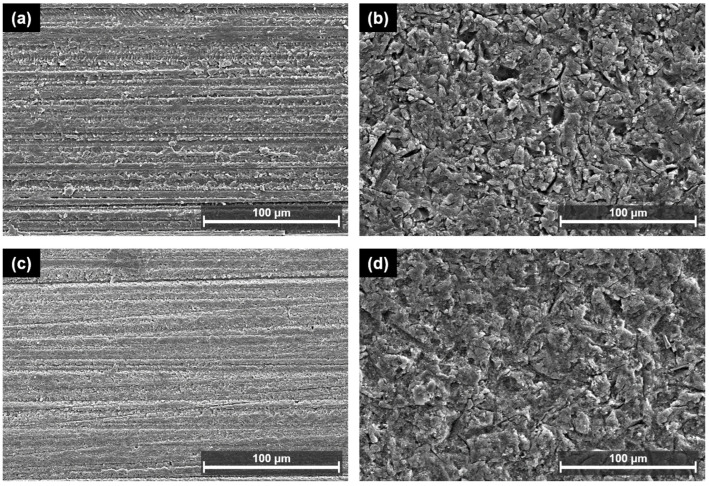
Low-magnification (×500) scanning electron microscopy (SEM) images of both types of 3D printing resins for different surface treatments: (**a**) No surface treatment of Group 1; (**b**) APA of Group 1; (**c**) no surface treatment of Group 2; and (**d**) APA of Group 2.

**Figure 7 polymers-15-01390-f007:**
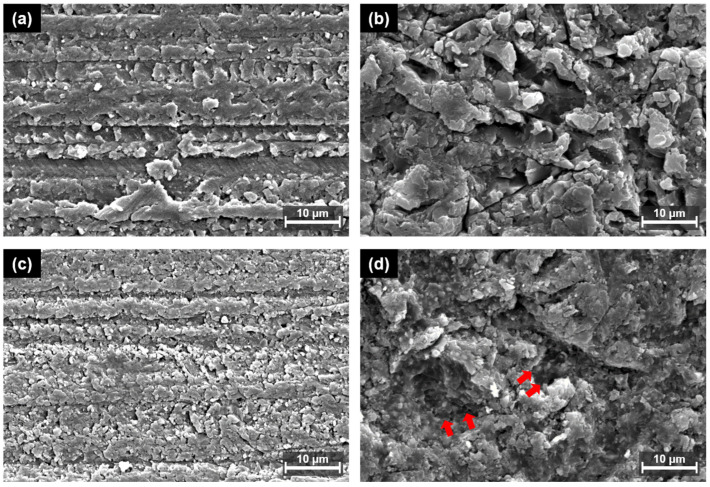
High-magnification (×2000) SEM images of both types of 3D printing resins for different surface treatments: (**a**) No surface treatment of Group 1; (**b**) APA of Group 1; (**c**) no surface treatment of Group 2; and (**d**) APA of Group 2. The red arrows indicate the deep, and wide porous surface.

**Figure 8 polymers-15-01390-f008:**
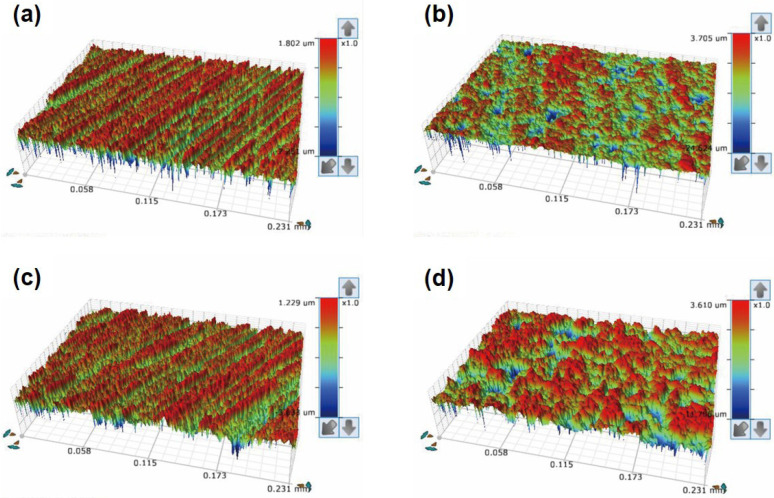
Representative 3D surface morphology images of both types of 3D printing resins for different surface treatments: (**a**) No surface treatment of Group 1; (**b**) APA of Group 1; (**c**) no surface treatment of Group 2; and (**d**) APA of Group 2.

**Table 1 polymers-15-01390-t001:** Composition of materials used in this study.

Product Name	Manufacturer	Composition
Tera Harz TC-80DP (A2)	Graphy, Seoul, Korea	Urethane dimethacrylate-based dental resin, phosphine oxides, and pigment
Permanent Crown (A2)	Formlabs, Somerville, MA, USA	Esterification products of 4,4′-isopropylidiphenol, ethoxylated and 2-methylprop-2enoic acid; ethoxylated bisphenol A dimethacrylate (Bis-EMA, methacrylate polymer), silanized dental glass, methyl benzoylformate, diphenyl (2,4,6-trimethylbenzoyl) phosphine oxide (TPO, photoinitiator), 30–50 wt.%—inorganic fillers (particle size 0.7 μm)
Single Bond Universal	3M ESPE, St. Paul, MN, USA	MDP, dimethacrylate resins, HEMA, Vitrebond copolymer, filler, ethanol, water, initiators, and silane
RelyX U200	3M ESPE, St. Paul, MN, USA	Base: methacrylate monomers containing phosphoric acid groups, methacrylate monomers, silanated fillers, initiator components, stabilizers, and rheological additives Catalyst: methacrylate monomers, alkaline fillers, silanated fillers, initiator components, stabilizers, pigments, and rheological additives
RelyX Ultimate	3M ESPE, St. Paul, MN, USA	Base: methacrylate monomers, silanated fillers, initiator components, stabilizers, rheological additives Catalyst: methacrylate monomers, alkaline fillers, silanated fillers, initiator components, stabilizers, pigments, rheological additives, fluorescence dye, and dark cure activator for Scotchbond Universal adhesive
Cobra Aluoxyd	Renfert GmbH, Hilzinger, Germany	50 µm aluminum oxide

**Table 2 polymers-15-01390-t002:** Description of study groups and respective adhesion conditions.

Group	Procedure
CU	Control + U200
AU	APA + U200
CBU	Control + SBU + U200
ABU	APA + SBU + U200
CBUT	Control + SBU + Ultimate
ABUT	APA + SBU + Ultimate

Control (no surface treatment), APA (airborne-particle abrasion using a sandblaster at a particle size of 50 µm and a pressure of 2 bar), and SBU (adhesive bonding agent, Single Bond Universal).

**Table 3 polymers-15-01390-t003:** SBS of both types of 3D printing resins before and after thermocycling.

		CU	AU	CBU	ABU	CBUT	ABUT
Group 1	Before thermocycling	17.0 ± 3.0 ^A^	20.8 ± 5.7 ^A^	22.9 ± 4.2 ^A^	24.7 ± 4.9 ^A^	21.9 ± 4.9 ^A^	20.2 ± 5.2 ^A^
	After thermocycling	8.1 ± 3.1 ^B^	16.9 ± 4.3 ^A^	11.6 ± 3.6 ^B^	22.5 ± 3.3 ^A^	16.8 ± 3.3 ^B^	17.8 ± 3.9 ^A^
Group 2	Before thermocycling	17.9 ± 2.8 ^A^	18.2 ± 2.6 ^A^	15.9 ± 2.6 ^A^	16.0 ± 2.8 ^A^	17.3 ± 2.4 ^A^	19.4 ± 2.7 ^A^
	After thermocycling	12.9 ± 2.3 ^B^	11.2 ± 2.6 ^B^	10.8 ± 1.8 ^B^	11.2 ± 2.6 ^B^	11.2 ± 2.3 ^B^	12.9 ± 3.0 ^B^

Uppercase letters indicate differences before and after thermocycling.

**Table 4 polymers-15-01390-t004:** Ra values of both types of 3D printing resins for different surface treatments.

	Control	APA
Group 1	0.41 ± 0.04 ^A,a^	0.79 ± 0.05 ^A,b^
Group 2	0.40 ± 0.04 ^A,a^	1.00 ± 0.06 ^B,b^

Uppercase letters indicate differences between material groups, and lowercase letters indicate differences between surface treatment groups.

## Data Availability

Not applicable.

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
