# Peer review of "Effect of Adhesion Conditions on the Shear Bond Strength of 3D Printing Resins after Thermocycling Used for Definitive Prosthesis"

_polymers, 2023, doi:10.3390/polym15061390_

Round 1

Reviewer 1 Report

The research is interesting and well conducted. The design of the study is coherent with the aim of the work ; the focus of the study is actual and also useful for the clinicians. The topic of the article is very technical and specific but has a lot of interesting points and cues from a clinical point of view.

1. The manuscript is clear and in my opinion relevant for the field it concerns.

2. The references are a lot but all recent.

3. The experimental design is well constructed and I think is useful also from a clinical point of view.

4. The results of the study are replicaple and are coherent with the materials and methods; I think that the results obtained demonstrate the validity of the study and its clinical role for future utilization of these materials.

5. The figures and tables are quite clear even though that the subject is quiet difficult to explain and interpretate. Even more, I found the tables very efficient for the readers.

6. The conclusions are consistent and in line with the evidence, arguments and also the hypothesis of the authors.

Author Response

First of all, thank you for considering my article for publication in Polymers. I am grateful to you for the valuable suggestions provided. English editing of the manuscript has been performed again, and the reviewer comments were provided in black color, whereas our responses are given in red. The appropriate changes made in the revised manuscript are highlighted. Please find our responses to the Reviewers’ and the Assistant Editor’s comments and recommendations below.

Here are responses to the reviewer comments:

- Thank you for reviewing our manuscript. We hope that our research will be useful clinically. We reviewed and revised the English language and style based on the reviewer's opinion.

Reviewer 2 Report

The presented paper describes effects of surface treatment on the adhesion between 3D printing resins used for the preparation of dental prosthesis and dental cement.  Several methods of surface preparation, 3D printable resins and cements are tested.  The interfacial adhesion strength between the resin and cement layers is measured.  Additionally, the thermal stability of the bond is characterised by measuring the interfacial adhesion strength following temperature cycling.

Overall, the standard of English is adequate.  The work is scientifically sound and tests useful properties of materials in the dental application space.  The more extensive use of 3D printing in this area will require such studies to be undertaken to understand the applicability of different materials in combination.  

However, there are some aspects of the 3D printing process which should be considered during this piece of work.  3D printing is an additive process, where a shape is built layer by layer.  This results in a 3D shape built up of many 2D layers.  This construction leads to some important considerations. The surface finish achieved will depend very much on the orientation of the part to the build platform of the 3D printer; surfaces parallel to the platform will be relatively smooth as they are a single layer, where surfaces perpendicular to the platform will be relatively rougher and will exhibit layer lines and artefacts from the build process.  Curved surfaces or surfaces angled between perpendicular and parallel to the plate will have surface finishes between these two extremes.

With this in mind:

·      Within the paper the authors only briefly describe the 3D printed shape dimensions and give no information on its actual geometry or how it was printed in terms of layer height or orientation on the build plate.  This information should be included to give a picture of the parts being tested, and of the starting surfaces before treatment.  The starting surface would be very different if the part was built parallel to the plate vs vertically.

·      Extending this, it would be very interesting, and ultimately vital to the applicability of these findings to understand if build orientation has an influence on these results. 

·      A comment should be made on the applicability of the work to parts built in different orientations.  This is important as the end goal of producing dental implants will result in the production of complex 3D geometries on which complex surfaces may be present.

·      Have the authors considered the degree of cure achieved during the printing and post curing process? This may be specified by the resin manufacturer but will likely affect the mechanical properties of the resin and therefore the impact of any mechanical surface treatments.  The degree of cure may also impact on the likelihood or possibility of any interlayer mixing occurring, so should be considered.  Photorheological data may be available for the resins used, or alternatively ATR-IR, looking at the consumption of allyl groups can also provide a measure.

·      The focus of the paper here is primarily on mechanical adhesion to the surface, although chemical modification of the surface is studied somewhat too with the use of SBU.  Further characterisation of the surface of the resins in terms of hydrophilicity / hydrophobicity may also be interesting to consider, as interfacial adhesion in terms of the surface chemistry is important.

·      While SEM of the surface as presented is useful and interesting, I think some lower magnification SEMs or optical microscopy would also be interesting as the length scale of feature that may be present from the printing process are more likely to be 50-100 ?m rather than <10 ?m

Overall, with consideration of the above points and addressing them in the text where appropriate, I would recommend the work for publication.

Author Response

First of all, thank you for considering my article for publication in Polymers. I am grateful to you for the valuable suggestions provided. English editing of the manuscript has been performed again, and the reviewer comments were provided in black color, whereas our responses are given in red. The appropriate changes made in the revised manuscript are highlighted. Please find our responses to the Reviewers’ and the Assistant Editor’s comments and recommendations below.

Here are responses to the reviewer comments:

Point 1

Within the paper the authors only briefly describe the 3D printed shape dimensions and give no information on its actual geometry or how it was printed in terms of layer height or orientation on the build plate.  This information should be included to give a picture of the parts being tested, and of the starting surfaces before treatment.  The starting surface would be very different if the part was built parallel to the plate vs vertically.

Response 1

The specimens used in this study were all printed in a direction parallel to the plate, and the layer thickness was set to 100 μm. The specimen used in this study has a smooth surface because it was printed parallel to the build plate. However, to ensure that specimen surfaces are uniformly controlled, we polished them with 1000-grit silicon–carbon abrasive paper.

Point 2

Extending this, it would be very interesting, and ultimately vital to the applicability of these findings to understand if build orientation has an influence on these results.

Response 2

It is expected that the build orientation may affect shear bond strength properties. This is because layers are formed on the surface depending on the build orientation. In this study, we did not observe effects of the build orientation on the shear bond strength because all specimens were printed in parallel to the build plate and, uniform surfaces were subsequently obtained by polishing. Therefore, as pointed out by the Reviewer, we agree that it is necessary to investigate adhesion conditions and strength according to build orientation. As a follow-up study, it is expected that the bonding conditions and methods used in actual clinical practice can be established by conducting a bond strength study using crown-shaped specimens.

Point 3

A comment should be made on the applicability of the work to parts built in different orientations.  This is important as the end goal of producing dental implants will result in the production of complex 3D geometries on which complex surfaces may be present.

Response 3

Because dental prosthesis has a very complex structure, it can be assumed that the surface will vary depending on the build orientation. Although the surface of the specimen was treated uniformly in this study, treating the internal surface of a prosthesis uniformly is expected to be difficult in clinical practice. Therefore, research should be conducted to derive the optimal printing and adhesion method that can be clinically applied by measuring the mechanical strength and bonding strength of crown-shaped specimens according to the build angle.

Point 4

Have the authors considered the degree of cure achieved during the printing and post curing process? This may be specified by the resin manufacturer but will likely affect the mechanical properties of the resin and therefore the impact of any mechanical surface treatments.  The degree of cure may also impact on the likelihood or possibility of any interlayer mixing occurring, so should be considered.  Photorheological data may be available for the resins used, or alternatively ATR-IR, looking at the consumption of allyl groups can also provide a measure.

Response 4

Because the degree of cure can affect the mechanical properties and surface treatment, it is considered very important to check the degree of cure by changing the post curing process. However, in this study, we used different surface conditions and cement after curing for each material according to the manufacturer's instructions. Therefore, the degree of cure was not confirmed after curing. Although many studies have investigated mechanical properties under various post curing conditions, not many have focused on adhesive strength. Therefore, it is considered important to conduct a study on the effect of surface treatment according to the degree of cure.

Point 5

The focus of the paper here is primarily on mechanical adhesion to the surface, although chemical modification of the surface is studied somewhat too with the use of SBU.  Further characterisation of the surface of the resins in terms of hydrophilicity / hydrophobicity may also be interesting to consider, as interfacial adhesion in terms of the surface chemistry is important.

Response 5

Mechanical surface treatment with APA can improve micromechanical retention by increasing the surface area and surface energy of a specimen. It can also maximize the wettability of silanes and resin cements by reducing surface tension. This results in a thin and deep interface when hydrophilic SBU is applied, and after air drying and light curing, SBU changes to hydrophobicity, increasing mechanical and chemical adhesion with resin cement. This is presented in the fourth paragraph of the discussion.

Reference:

  1. Kwon, S.-J.; Park, J.-K.; Son. S.-A. Shear bond strength between universal adhesives with various pH and dual-cured resin cements. Kor. J. Dent. Mater. 2018, 45, 301–309. DOI: 10.14815/kjdm.2018.45.4.301

Point 6

While SEM of the surface as presented is useful and interesting, I think some lower magnification SEMs or optical microscopy would also be interesting as the length scale of feature that may be present from the printing process are more likely to be 50-100 ?m rather than <10 ?m

Response 6

In this study, ×2000 high-magnification SEM images are presented. Additionally, low-magnification images were obtained and are presented in Figure 6. These images were measured at ×500 magnification with the length scale of 100 μm, and the surface characteristics of each material can be confirmed even at low magnification.
